# IN DEFENSE OF THE PAPER

**Owen Lockwood**
Department of Computer Science
Rensselaer Polytechnic Institute
Troy, NY 12180, USA
`lockwo@rpi.edu`

## ABSTRACT

The machine learning publication process is broken, of that there can be no doubt. Many of these flaws are attributed to the current workflow: LaTeX to PDF to reviewers to camera ready PDF. This has understandably resulted in the desire for new forms of publications; ones that can increase inclusively, accessibility and pedagogical strength. However, this venture fails to address the origins of these inadequacies in the contemporary paper workflow. The paper, being the basic unit of academic research, is merely how problems in the publication and research ecosystem manifest; but is not itself responsible for them. Not only will simply replacing or augmenting papers with different formats not fix existing problems; when used as a band-aid without systemic changes, will likely exacerbate the existing inequities. In this work, we argue that the root cause of hindrances in the accessibility of machine learning research lies not in the paper workflow but within the misaligned incentives behind the publishing and research processes. We discuss these problems and argue that the paper is the optimal workflow. We also highlight some potential solutions for the incentivization problems.

## 1 INTRODUCTION

Machine Learning is one of the most popular fields of research, with tens of thousands of papers being produced every year[1]. Contemporary machine learning research is dominated by deep neural networks (LeCun et al., 2015), which have achieved well publicized successes in a variety of fields including (but not limited to): Computer Vision (Krizhevsky et al., 2012), (Redmon et al., 2016), Generative Modelling (Goodfellow et al., 2014), (Karras et al., 2019), Natural Language Processing (Sutskever et al., 2014), (Devlin et al., 2018), and Reinforcement Learning (Mnih et al., 2015), (Silver et al., 2016), (Vinyals et al., 2019). Parallel to this explosion in research volume are the potentially problematic implications of this research. These range from privacy and discrimination concerns of large language models (Carlini et al., 2020), (McGuffie & Newhouse, 2020), (Bender et al., 2021) to environmental effects (Strubell et al., 2019), to algorithmic bias (Buolamwini & Gebru, 2018), (Hooker et al., 2020), (Bagdasaryan & Shmatikov, 2019), and further ethical implications of widespread adoption of machine learning systems (Aïvodji et al., 2019). These are important works concerned with the greater social implications of machine learning research, which differs from the focus of this work: the machine learning research process itself.

When evaluating the publication system as a whole, not only can the existing problems be explained, but we can begin to understand the mechanisms of why machine learning research has these problems. While the symptoms of systemic problems are often discussed, e.g. the reproducibility crisis (Pineau et al., 2020), hypothesizing after experiments (Gencoglu et al., 2019), etc. the root cause of these issues are often ignored or misattributed. Misaligned incentives were previously mentioned by Sculley et al. (2018); however, it was only touched on briefly and their solutions did not address any systemic incentivization problems. Lipton & Steinhardt (2018) provides an excellent overview of existing problems in machine learning publications, but offers little in terms of the origin of these problems (once again only briefly mentioning misaligned incentives) or what to do about them.

---

[1]https://arxiv.org/help/stats/2020_by_area/index

The rationale for this workshop is understandable and important, contemporary machine learning papers are often far from the pinnacle of inclusivity, explainability, and interpretability. The idea that a new conceptualization of the research publication will help alleviate some of these problems is a natural one. However, it fails to consider the problem as a whole, missing the forest for the trees. These new formats will fail to meaningfully solve any of the existing problems in accessibility and will likely decrease inclusivity because they will not address any of the primary root causes of these problems.

## 2 MISALIGNED INCENTIVES

Here we highlight the problems that exist in the publication system that decrease accessibility, explainability and interpretability. These are often attributed to the paper workflow; however, we argue that these problems are the result of misaligned incentives that merely present themselves in the paper (the basic unit of contemporary research).

### 2.1 DEANS DON'T READ, THEY ONLY COUNT

The first problem is one that decreases inclusivity in the research production process. The expected pace of producing research, especially for graduate students and early career researchers, is a major problem. Not only is this pace of research extremely exclusive, those able to keep up with it pay the price in mental stability (Chirikov et al., 2020). This problem has only been exacerbated by the COVID-19 pandemic (Gewin, 2021). There are usually 5 major conferences per year that publications are expected in: Association for the Advancement of Artificial Intelligence (AAAI) Conference on Artificial Intelligence, Conference on Neural Information Processing Systems (NeurIPS), International Conference on Machine Learning (ICML) with the other two being subfield specific. E.g. for Natural Language Processing: Association for Computational Linguistics (ACL) Conference, Empirical Methods in Natural Language Processing (EMNLP), for Reinforcement Learning: International Conference on Learning Representations (ICLR), for Computer Vision: Conference on Computer Vision and Pattern Recognition (CVPR), European Conference on Computer Vision (ECCV), International Conference on Computer Vision (ICCV), etc. We list all of these to emphasize the magnitude of the publishing expectations. How does this problem manifest itself in papers? Via the (in)famous concept of the "Salami Publication" (Šupak Smolčić, 2013), also called the Minimum Publishable Unit (MPU). The MPU is exactly what the name indicates: the smallest amount of work/research that can result in a publication. Given the publication expectations, why would one publish a "better" paper in one venue, when one could publish two lower quality papers in two venues? If you could get one paper in NeurIPS or one paper in AAAI and NeurIPS, one would always chose the latter. This isn't meant to blame the researchers, it is exactly what a rational actor would do. Existing incentives (e.g. funding, promotions, academic/social capital, etc.) rewards the number of publications and having a publication in the major conferences, something that can be objectively measured, not the quality of papers, something that cannot be objectively determined. This inevitably pushes researchers towards the MPU (Edwards & Roy, 2017). While this is undoubtedly problematic and is a considerable source of exclusionary practices in research production, it is not attributable to the paper workflow. Although we can (subjectively) observe papers becoming more "salami", it would be mistaken to assume that this is a result of the paper. It is merely because the paper is the basic unit of research expression, thus flaws in the system will naturally express themselves through it.

### 2.2 WINNING ISN'T EVERYTHING, IT'S THE ONLY THING

The second problem is the constant push for State of the Art (SotA) results on a set of standard benchmarks. It is generally understood in research that when presenting an empirical method, SotA results are important to getting published. Thus, this incentives SotA-hacking, the machine learning equivalent to p-hacking (Head et al., 2015). This limits the explainability of papers and is a severe pedagogical weakness. To understand why this is the case, we first need to evaluate the results of this incentivization. Because SotA results are key to successful publications, it quickly became the goal of machine learning research; a publication version of Goodhart's law[2] (Goodhart, 1984). When

---

[2]"When a measure becomes a target, it ceases to be a good measure"

the goal is SotA results, as opposed to thorough and impactful contributions to human knowledge, it results in poor scientific practices. One such example is Hypothesizing After Results Are Known (HARKing) (Kerr, 1998): conducting experiments and once you have empirical results that are publishable, working backwards to fill in why this technique works. A common variation of HARKing in machine learning research is called "Grad Student Descent" (Gencoglu et al., 2019), i.e. running experiments in which you change hyperparameters (largely at random) that result in a mild increase in performance which is then (mis)attributed to a seemingly substantial theoretical change. This backfill is necessary because random hyperparameter changes that result in SotA results is not a publishable methodology. This misattribution of the source of improvements is a severe problem for explainability. It is not uncommon for SotA results to be attributed to algorithmic differences, when the real origin of the improvement is something far more banal, such as random seeds (Henderson et al., 2018) or code level changes (e.g. learning rate, gradient clipping, etc.) (Engstrom et al., 2020), (Tucker et al., 2018). This SotA misincentivation manifests itself via the paper, resulting in publications that are severely flawed and obfuscatory (to hide the true origin of their performance gains). Here, again, we can see it is easy to blame the paper for lacking explainability; however, this is not a result of the paper workflow, but the incentives behind it.

## 2.3    Don't Hate The Player, Hate The Game

A third problem in machine learning research publications is the theoretical/mathematical foundations, or lack-thereof. The meaningless (and not infrequently incorrect or irrelevant) mathematics and equation packing included in machine learning papers is a severe hindrance to the interpretability of papers. The general form of many research papers is: introduction, background, theoretical (mathematical) foundation, empirical results, conclusion. This is not inherently a bad way to structure a paper. The problem is that because there is an expectation for a thorough mathematical background, papers that do not naturally fit this format must mutilate themselves into it. Even papers that do fit this format are not immune mathematical flaws (Reddi et al., 2019). The faulty incentives here act on both reviewer and researcher. Reviewer incentives encourage acceptance of papers that conform to the *status quo* of the existing normal science paradigm (Kuhn, 2012), expecting extensive mathematical/theoretical justifications for the empirical results. Researcher incentives encourage publications, leading to the reviewers being seen as an obstacle (rather than an important part of the process) and can be subject to animosity (Peterson, 2020). Rather than reviewer and researcher sharing the same goal of improving research, sharing important work with the scientific community, making papers as understandable and interpretable as possible, their incentives encourage the opposite. These expectations result in mathematical foundations that are intentionally long and obfuscatory, often irrelevant and, sometimes, outright wrong (Lipton & Steinhardt, 2018). What could be said in a few clear and coherent sentences is said in unhelpful lemmas and proof-filled appendices (note this is not a complaint of math in papers, but of *unnecessary* math in papers that hurt interpretability). Typically, this math offers some "intuition" into why the method works, but offers little in terms of mathematical rigour or relevance. This intuition often comes in the form of proofs of error bounds or convergence properties, which may be mathematically valid, but are irrelevant when their assumptions get thrown out the window because of the usage of massively non-convex deep neural networks. Consider the famous reinforcement learning paper Haarnoja et al. (2018)[3], in which convergence proofs are given for tabular soft policy iteration (taking up nearly a page) but none of the proofs actually apply to their algorithm which uses neural networks. This technique of proving the tabular case but using neural networks for the main algorithm (making the tabular proofs largely moot) is far from rare in RL research (Lan et al., 2020), (Dabney et al., 2018). This is problematic because this "intuition" is often wrong when it comes to neural networks. It is known that Q-learning (Watkins & Dayan, 1992) has guaranteed convergence properties; but when a neural network is used, these properties become meaningless (convergence being one of the largest problems of Deep Q Networks) (Mnih et al., 2013). Once again, when reading contemporary research and seeing this meaningless math, it is easy to blame this on the paper. But the uninterpretable mess is not a result of the paper process, but the incentivization behind it.

---

[3]This is not meant as a slight against that work, as it was produced within the aforementioned incentivization structure

## 3 WHY NOT CHANGE?

It is here that we would like to offer a rebuttal to the work being submitted to this workshop. While the submissions will surely be high quality pieces of research, they will ultimately fail to solve any of the problems previously laid out and fail to meet the desires goals of increasing inclusivity, explainability, and interpretability. This seems like a bold claim, given that we cannot see what has been submitted; however, we are able to make it because ultimately any change to the paper workflow is just that: a workflow change. Any change that doesn't fundamentally alter the publication process, instituting systemic changes, cannot succeed in addressing the primary concerns of this workshop.

### 3.1 THE DOUBLE-EDGED SWORD

Let us first consider what we argue is the tradeoff of inclusivity vs explainability. Increasing inclusivity requires that the machine learning research field is more open to those contributing to research (especially to underrepresented/disadvantaged groups). However, increasing the pedagogical strength of work can act to counteract inclusivity efforts. First, consider the existing example of distill.pub. These works represent a step up in pedagogical strength (increasing explainability and visualization). However, this requires substantially more time and skills than traditional publications (as the journal notes). We cannot increase inclusivity in research while also increasing the number of (non-research) skills required for a publication. Other solutions that have better visualizations inevitably increase barriers, as machine learning research is not about visualizations (thus requiring additional skills). If someone has done research worth publishing, but there are barriers to publication (visualization, advanced explainability techniques, etc.), it limits both the researcher and the community. Here we can see the paper as the optimal form of publication. The paper represents the lowest entry requirements when it comes to inclusivity. If one has completed some research, writing it down represents the lowest conceivable barrier (the bare minimum of communication). It requires no further technical or research skills, only the ability to write (which is an implicit prerequisite for machine learning research). Any other publication workflow increases this barrier. Whatever the reconceptualization is, it is definitionally not the paper and therefore will require more skills (increasing the barrier), be in in web design, visualization, rendering, video communication etc. Not only is the paper optimally inclusive (lowest barrier to entry) it also has an extremely high explainability ceiling. Papers have the ability to be extremely effective methods of communicating machine learning research. The fact that the paper also has a low explainability floor, resulting in many papers that aren't very well written does not mean it is an ineffective format, merely that the incentives need to be changed to reward explainability.

One might, understandably, challenge this and say something along the lines of, "we would never require every single paper to instantly transition to our new style and use [whatever new techniques are introduced]. The traditional paper workflow will still exist, we are merely augmenting it". However, this once again fails to address the fundamental problems causing the inaccessibility. Augmenting the paper workflow sounds promising but even that would decrease inclusivity if the other problems are not fixed. Without changing the sheer number of publications, these "augmented" publications would naturally rise to the top (as existing publications from prominent labs/groups already do). When reading (and citing) research, we reach not the work that has the most inclusive practices, but the easiest to cite and read. Second, we can see that, in reality, most of these augmented publications would change little. There are already existing "augmented" publications in the form of blogs and talks and demos that go along with some papers. These are undoubtedly improving the visualization and the communication of complex ideas. However, the same groups that currently use these techniques would also be the only groups to use the new publication methods as well: large companies and academic labs. This offers little in the way of increasing inclusivity.

### 3.2 EXPLAIN, BUT NOT TOO MUCH

What exactly is the purpose of publications? One of the core questions that this workshop asks is: "How do we communicate ML research and theory more effectively?" As we previously discussed, there is no doubt a problem with explainability in contemporary papers. However, it is important to recognize that ultimate goal of a research publication is to convey the information to a (relatively) specialized audience. While scientific communication to larger audiences/the general public is important, that is not the role of the publication. Therefore, when seeking to improve explainability,

it is important to remember the audience (i.e. who we are explaining to). There needs to be some expectation of background knowledge in order to effectively and concisely convey research to ones peers.

## 4 WHAT IS MACHINE LEARNING?

Before getting to potential solutions, we will briefly touch on an important philosophical concept that often goes undiscussed in machine learning research. That is the question of the epistemological basis of machine learning, or given the crossroads we are at as a field, what the basis of machine learning research *ought* to be. Should what much of the field is doing be part of the publication process? As we previously discussed, much of machine learning research is defined by a hyper focus on benchmarks (Wagstaff, 2012). While random hyperparameter optimization isn't a theoretically sound method of research, that doesn't mean it isn't of any use. Better hyperparameters can help with machine learning models used in real world applications. Even if incentives are fixed, these results are still important, but they perhaps there is a better place for them than the proceedings of conferences.

When asking what machine learning should be, we have to consider what the goal of machine learning research is. Machine learning is in a unique position, given it's unusually strong ties to industry. At NeurIPS'20, a single company (Google) was involved in more than 12% of all papers[4]. While corporations can contribute meaningful research to scientific fields, the substantial presence begs the question: is machine learning research's goal really a scientific one? While the "goal of science" is difficult to philosophically define and there is a long history of debate (Leplin, 1984), we are able to sidestep that dilemma by simply contrasting it with the goal of what we call the "industrial paradigm". Industrial paradigm machine learning research attempts to improve techniques used in practice, without concern for fundamental knowledge. This breakdown might, at first, sound like a simple division of theory vs. applied work. However, the focus of the industrial paradigm is more specific than applied work. It isn't just about applying machine learning to a real world problem, it's about applying machine learning to an actual corporate problems, the solution to which results in increased revenue for the company and has little further implications. Mnih et al. (2015) represents an example of applied research, done in an industry lab, but is not within the industrial paradigm. This also represents the idea that industrial research is not just research that comes from corporate labs and can originate in any lab. This paradigm can also help explain machine learning's atychiphobia (fear of negative results). Negative results are important contributions in science, with a number of scientific journals dedicated purely to them (e.g. the Journal of Negative Results collection). However, negative result publications have been dropping (even before the machine learning research boom) (Fanelli, 2012). Negative results are substantially less important to industrial research (given the focus on commercial solutions); hence, negative machine learning publications are not just less common, but almost unheard of. While this industrial paradigm of research may occupy a sizeable fraction of publication (even with incentive changes) due to the nature of industry involvement, it is important to consider whether this is what we, as a field, want.

## 5 SOLUTIONS

We highlight how existing solutions fit within our incentivization model and how they can be expanded further, in addition to providing some of our own solutions.

### 5.1 SHORT TERM SOLUTIONS

What can be done about these problems today? Institutional, large-scale changes inevitably require significant time. While we want to enact these changes, we also want to increase accessibility in near term research. There are several steps we can take to increase accessibility for near term research (many of which are already underway). None of these changes require any modification to the current paper workflow and will be able to meet the goals better than any workflow modification. One example, that the world was forced into, is online conferences. For post-pandemic conferences maintaining hybrid options can enable those without the ability to travel contribute to the research

---

[4]Data Hyperlink

at a conference. This supports the globalization of research and underfunded researchers. Another important step is the democratization of computational resources. Even with advancements in computational power and techniques, deep learning research is not very accessible to those lacking resources. In the entire continent of Africa, there exists only one supercomputer on the Top 500 list (the Toubkal)[5]. Government and corporate funding is essential to increase the accessibility of computational resources. This is might seem like a lofty goal, but it is entirely doable in the immediate future because it doesn't require any value realignment of funding agencies. I.e. we do not have to convince any government or corporation to act more egalitarian, we just have to show how it is in their rational self interest to increase funding to these potentially massive sources of research[6]. These are just a few ideas; however, these near term steps are not the focus of this paper as they aren't making any changes to the incentives.

## 5.2 Long Term Solutions

Although the short term solutions are promising, we need substantial changes to machine learning research incentives. Here we address each of the misaligned incentives previously discussed and present potential solutions, showing how the paper workflow can be used to increase accessibility.

The first incentive discussed was the problem of "publish or perish". Specifically, the expectation is an unreasonably large number of publications that results in more, lower quality, publications. It is clear that we need change, and while this is a common sentiment (Bengio, 2020), actual policies are far less common. The reason policies are so rare is because most try to address the actual publication cycle, or try to change something about the publication process. However, this fails to address the real problem, which is that there is the *expectation* to publish is so many conferences. Therefore, the only effective policy will be one that changes this expectation. One example is a cultural change that has been promoted for some time now: the concept of "Slow Science" (Lutz, 2012), (Stengers, 2018). The idea of Slow Science is to encourage cultural shifts towards publishing slower, more impactful publications, which would enable research to flourish when it would otherwise be stifled by the pace of contemporary publications (Aitkenhead, 2013). One could also consider adjustments to the reward systems for publications. The number of publications should be significantly less considered when determining funding, academic capital, etc. One could also consider a more radical maneuver of greatly reducing or ending the large (non-specialized) ML conferences (NeurIPS, AAAI, ICML), thus eliminating the possibility of the unhealthy publication expectations. Given the number of papers in these conferences, it is not feasible to engage with (much less read) even a small fraction of the accepted works. If these massive conferences just naturally fracture into the different specializations (NLP, RL, CV, theory, etc.) why not just have smaller more niche conferences? This would help to reduce the publication drive (by reducing the number of conferences) and could make the conferences better at achieving their goals of scientific communication. These maintain the paper workflow, but increase accessibility by changing the incentives such that the number of publications expected is decreased.

The second problem is the SotA expectation. The incentive for SotA-hacking stems from the accepted knowledge that SotA results in publications. If you want to get published in big conferences with empirical work, SotA results are expected. While reducing the publication demand will help with this problem, it is not a complete solution. In addition, the conference/publication expectations need to change. This will likely be much easier to change than the previous incentive, because it is merely a result of the cultural norms rather than any actual regulations. On the specific problem of HARKing, one potential solution is to require hypothesises prior to experiments being conducted, requiring well defined ideas rather than random hyperparameter tuning. This is something that has already seen some interest[7]. These shifts will result in better pedagogical approaches, as the SotA requirements are relaxed enabling better science to be conducted. This will enable more explainable papers, as researcher's starting point will not longer be SotA results, but proper scientific investigations, eliminating the need to backfill.

---

[5]www.top500.org/lists/top500/2020/11/

[6]As Gould (2010) put it, "I am, somehow, less interested in the weight and convolutions of Einstein's brain than in the near certainty that people of equal talent have lived and died in cotton fields and sweatshops."

[7]The Pre-Registration Experiment NeurIPS'20 Workshop

The third concern has to do with reviewers. Specifically, the incentives for reviewers are broken. Reviewers expect mathematics (leading to equation packing) and yet do not check or verify the correctness of this math leading to well publicized instances of faulty mathematics (Lipton & Steinhardt, 2018). Simply put, there is almost no external incentive for reviewers to actually be good reviewers. The flaws of the review system has seen a fair amount of (unofficial, e.g. twitter) discourse. Some ideas include paying reviewers, creating an anonymous rating system of reviewers, eliminating double blind peer review and shifting towards a continuous arXiv review, etc. We have little to add to this topic, other than to note that this is an example of solutions that might actually work, because they address the root cause of the problem: the incentives. By changing review incentives so that papers are valued for their interpretability, not their ability to appear impressive behind a wall of incomprehensible math, will be of substantial benefit (and once again requires no paper process changes).

Finally, we want to briefly address a potential solution to the industrial paradigm. This may not universally be seen as a problem, but we believe that it has substantial limitations. Common in industry is the practice of patenting algorithmic developments. These patents are not conducive to open research. Machine learning patents are not patenting the software, but the ideas behind the algorithms. Examples include Google patenting "Methods ... for asynchronous deep reinforcement learning ... wherein each worker is configured to operate independently of each other worker ... each worker is associated with a respective actor that interacts ... the environment during the training of the deep neural network" (Mnih et al., 2021), "Methods ... for training an actor neural network ... methods includes obtaining a minibatch of experience tuples and updating current values of the parameters of the actor neural network ..." (Lillicrap et al., 2020), and Amazon patenting "a system for capturing and processing portions of a spoken utterance command that may occur before a wakeword" (Yasa et al., 2021). Patenting machine learning algorithms is representative of industrial research and in opposition with the global and open scientific initiatives. Expanding patent laws to further limit algorithmic patenting may help to curb these practices, something the Supreme Court of the United States has been trending towards. In Diamond vs. Diehr (1981) the ruling stated "a mathematical formula, like a law of nature, cannot be the subject of a patent" and SAP America vs. Investpic LLC (2018) stated that "mathematical calculations and formulas are not patent eligible". Fundamentally machine learning algorithms are automated mathematical calculations, so legally clarifying this may help to enforce open research practices.

## 6   CONCLUSION

Contemporary machine learning research is problematic. The way forward is not through temporary band-aids of publication reconceptualizations, but through thorough institutional and systematic changes. Fixing these incentives will enable machine learning research to become a more inclusive research space, enabling better pedagogy. We recognize that our solutions are limited, and we often highlighted existing techniques; however, we are hopeful that this work will spark further discussion on solutions to be reached as a community (as they ought to be). The main faulty incentives are that of research production expectations, SotA expectations and reviewer *status quo* expectations. All three of these reduces the accessibility of the paper workflow. However, any other publication form would be subject to the same flaws. We can directly incorporate the goals of inclusivity, explainability, and interpretability within the publication process thus resulting in increased accessibility, all within the paper process.

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

Ravi Chandra Reddy Yasa, Sai Rahul Reddy Pulikunta, Eliav Kahan, and Gregory Newell. Generating input alternatives, March 18 2021. US Patent App. 17/109,449.

