# OpenReview forum: "In Defense of the Paper"
_ICLR.cc/2021/Workshop/Rethinking_ML_Papers — Rethinking ML Papers - ICLR 2021 workshop Oral_

### Official Review · AnonReviewer1 · 2021-03-29
**Good compilation of the agruments for the current PDF format**

**Accessibility:**

Score of 4 (Strong): Submission states accessibility concerns and provides solutions within the proposed framework. However, it does not declare the limitations and exceptions.

**Litreview:**

Score of 4 (Strong): The submission directly differentiates itself from previous works and formats.

**Problemstatement:**

Score of 4 (Strong): The submission sets a very strong example of how to address the problem, which should be relevant to the workshop themes.

**Relevance:**

Score of 4 (Strong): The submission directly addresses a theme of the workshop, and does so in a very professional manner.

**Results:**

Score of 4 (Strong): Submission is very well structured and follows all the criteria (i.e. clarity, novelty, interactivity, and coherency). However, practical significance/theoretical implications are not discussed.

**Reviewerconfidence:**

I see no reason why this paper could be NOT worth reading and discussing.

**Reviewtext:**

The paper defends the current way of publishing ML-results: PDF paper and argues that major changes in it are hurting more than helping. The two reasons are
- problems with the current workflow are not because of the publication format, but because of the expectations and incentives our field has. Specifically: sota-expectation, math-expectation, many-papers-expectation.
- more advanced publication formats, e.g. distill.pub would set an even higher bar and require even more resources, than we have now. E.g. in addition to writing a clear paper with a sota results (see an argument about expectation), one would need to also know about web-design, make animations, etc.

An additional statement, which I have not seen online much (section 3.1-3.2) and apparently like is that the scientific publication has a specific goal and does not necessarily need to be optimized for pedagogical purposes.  "increasing the pedagogical strength of work acts counteracts inclusivity efforts."


I do not agree with some of the statements, which the paper does, e.g. "Better hyperparameters can help with industrial and machine learning models used in the real world. Even if incentives are fixed, these results are still important" - yes. "but they need a place to exist outside of conference" - not necessarily.  I can see a happy world, where "hyperparameters track" exists in the conferences at a separate track, as the publications are the currency of academia and this would not likely to change.

The weak point of the paper is lack of proposed solutions. E.g "The number of publications should be significantly less considered
when determining funding, academic capital, etc." - yes, but how? Nevertheless it would be stupid to require a solution to a such global problem from the workshop paper.

I believe that community would strongly benefit from the discussion, which paper starts.


**Score:**

Strong accept: The reviewer has a strong enthusiasm to apply the proposed framework in their work.

---

### Official Review · AnonReviewer2 · 2021-03-30
**Interesting work**

**Accessibility:**

Score of 4 (Strong): Submission states accessibility concerns and provides solutions within the proposed framework. However, it does not declare the limitations and exceptions.

**Litreview:**

Score of 3 (Neutral): The submission acknowledges previous work, but does not necessarily explain how the submission differentiates itself (i.e we want to avoid the “deluge of citation” strategy, leaving the reviewer to click through references and figure this part out for themselves).

**Problemstatement:**

Score of 4 (Strong): The submission sets a very strong example of how to address the problem, which should be relevant to the workshop themes.

**Relevance:**

Score of 4 (Strong): The submission directly addresses a theme of the workshop, and does so in a very professional manner.

**Results:**

Score of 4 (Strong): Submission is very well structured and follows all the criteria (i.e. clarity, novelty, interactivity, and coherency). However, practical significance/theoretical implications are not discussed.

**Reviewerconfidence:**

4 - The reviewer is fairly confident about the evaluation

**Reviewtext:**

This paper identifies existing problems in machine learning research and argues that rather than identifying problems in the paper workflow, the incentives behind research need to be addressed which can lead to more meaningful and focused research.
 Some of the  problems highlighted in this work include publish or perish idealogy, SotA hacking,HARK etc.

The authors also present some short term and long term solutions such as Hybrid models for post-pandemic conferences, democratization of resources and the pre-registration of papers leading to streamlined research. The authors also correctly identify that research publications need to be addressed to a focused audience who possess some background knowledge rather than a more generalized audience. Increasing focus towards visibility and visualizations would lead to less inclusivity among researchers.

However, some of the solutions presented in the paper such as pre-registration and democratization of computational resources were already discussed in previous works. The idea of Slow Science is interesting but a more concrete framework on how this could be implemented is missing. There exists a tradeoff between impact of research work and number of publications, so a detailed discussion on the impact of these changes and whether resources such as funding need to have other metrics for decision making would be interesting.

The incentivization of reviewers using anonymous rating is another solution but more defined ideas on how this could be implemented and what changes are required would have been useful. Since this process should ideally be objective, a look at whether feedback from both authors and organizers need to considered or just the organizers would lead to better discussion.

Overall an interesting work that identifies problems in the community and research process, while presenting preliminary solutions. However, a clear plan and framework to implement these ideas would lead to tangible changes.

**Score:**

Accept: The reviewer believes the submission provides a novel and reliable scheme to improve science communication but needs improvement.

---

### Meta-Review · Area_Chair1 · 2021-04-01

**Recommendation:** Accept
**Confidence:** 5

**Metareview:**

In a workshop geared towards rethinking the scientific dissemination process in the ML community, this submission argues strongly in favour of current dissemination formats. In particular, the submission argues for a systemic change in the incentives in-place in the research ecosystem.

This paper raises several interesting discussion points---as both reviewers duly noted---including
i) newer, interactive publishing formats will demand more of the ML author
ii) merely rethinking publication "formats" wouldn't solve the underlying problem; we need to rethink the incentivization of publications and associated bibliometrics
iii) the primary audience of a publication are members of the scientific community;

The paper doesn't stop at raising these discussion points; it also provides potential solution sketches. Both reviewers flag a concern that the solutions aren't well fleshed out; despite that I think this paper provides enough food for thought. As one reviewer rightly points out,
> "I see no reason why this paper could be NOT worth reading and discussing."

As organizers of this workshop, we would be doing injustice to the goals of the workshop if we wouldn't welcome adversarial perspectives. In favor of an inclusive dialogue, I would like to strongly recommend this submission for an oral presentation at the workshop. I commend the author(s) for the interesting viewpoints put forth.

---

### Decision · Program_Chairs · 2021-04-01

Accept (Oral)